# Transient nuclear Prospero induces neural progenitor quiescence

**Sen-Lin Lai[1,2], Chris Q Doe[1,2]\***

[1]Institute of Neuroscience, Howard Hughes Medical Institute, University of Oregon, Eugene, United States; [2]Institute of Molecular Biology, University of Oregon, Eugene, United States

**Abstract** Stem cells can self-renew, differentiate, or enter quiescence. Understanding how stem cells switch between these states is highly relevant for stem cell-based therapeutics. *Drosophila* neural progenitors (neuroblasts) have been an excellent model for studying self-renewal and differentiation, but quiescence remains poorly understood. In this study, we show that when neuroblasts enter quiescence, the differentiation factor Prospero is transiently detected in the neuroblast nucleus, followed by the establishment of a unique molecular profile lacking most progenitor and differentiation markers. The pulse of low level nuclear Prospero precedes entry into neuroblast quiescence even when the timing of quiescence is advanced or delayed by changing temporal identity factors. Furthermore, loss of Prospero prevents entry into quiescence, whereas a pulse of low level nuclear Prospero can drive proliferating larval neuroblasts into quiescence. We propose that Prospero levels distinguish three progenitor fates: absent for self-renewal, low for quiescence, and high for differentiation.

## Introduction

Quiescent stem cells preserve the stem cell pool for activation following disease or injury. Moreover, cancer stem cells can enter remission during treatment, possibly using mechanisms similar to stem cell quiescence, and re-initiate proliferation months to years later. Thus, studying stem cell quiescence may help design clinical approaches to prevent tumor dormancy and cancer recurrence, as well as to aid in activating stem cells for tissue repair.

*Drosophila* neuroblasts have served as a model system for identifying conserved signaling pathways that regulate stem cell proliferation and quiescence (*Ebens et al., 1993*; *Datta, 1995*; *Britton and Edgar, 1998*; *Doe et al., 1998*; *Egger et al., 2008*; *Tsuji et al., 2008*; *Sousa-Nunes et al., 2010*; *Homem and Knoblich, 2012*; *Weng and Cohen, 2012*). *Drosophila* neuroblasts delaminate from neuroectoderm during early embryogenesis, and then they go through multiple rounds of asymmetric cell division before exiting cell cycle at the end of embryogenesis; upon cell cycle exit, neuroblasts can undergo apoptosis (*Karcavitch and Doe, 2005*; *Maurange and Gould, 2005*; *Ulvklo et al., 2012*) or enter quiescence (*Truman and Bate, 1988*; *Datta, 1995*; *Tsuji et al., 2008*; *Chell and Brand, 2010*; *Sousa-Nunes et al., 2011*). During neuroblast asymmetric division, the scaffolding protein Miranda (Mira) is partitioned into the differentiating daughter cell, called a ganglion mother cell (GMC), and it carries at least three cargo proteins: the translational repressors Staufen and Brain tumor (Brat) and the transcriptional repressor Prospero (*Hirata et al., 1995*; *Broadus and Doe, 1997*; *Ikeshima-Kataoka et al., 1997*; *Broadus et al., 1998*; *Schuldt et al., 1998*; *Shen et al., 1998*; *Bello et al., 2006*; *Betschinger et al., 2006*; *Lee et al., 2006*). In the GMC, Mira appears to be degraded and its cargo is released into the cytoplasm (Staufen, Brat) or the nucleus (Prospero). Prospero is an atypical homeodomain protein that directly binds and represses progenitor and cell cycle genes to initiate GMC and neuronal differentiation (*Li and Vaessin, 2000*; *Choksi et al., 2006*); the mammalian ortholog Prox1 has a similar role

**\*For correspondence:** cdoe@uoregon.edu

**Competing interests:** The authors declare that no competing interests exist.

**eLife digest** Stem cells provide tissues in the body with a continuing source of new cells, both when the tissues are first developing and when they are growing or repairing in adulthood. A stem cell can divide to create either another stem cell, or a cell that will mature into one of many different cell types.

Neuroblasts are a type of brain stem cell and can divide to create two new cells: another neuroblast that will continue to replicate itself and a cell called a ganglion mother cell that will go on to produce two mature cells for the nervous system. Moreover, when a neuroblast divides, it splits unequally, so that certain molecules end up predominantly in the ganglion mother cell—including a protein called Prospero. Once partitioned into the ganglion mother cell, the Prospero protein enters the nucleus, where it switches off 'stem cell genes' and switches on 'neuron genes' so the ganglion mother cell can form the mature neurons of the brain. Thus, neuroblasts must keep Prospero out of the nucleus to maintain stem cell properties, whereas ganglion mother cells must move Prospero into the nucleus to form neurons.

Now, Lai and Doe discover a new way that the Prospero protein is used to control stem cell biology. Neuroblasts, like all stem cells, can enter periods where they go dormant or quiescent—that is, they temporarily stop generating ganglion mother cells. By analyzing which proteins are present in neuroblasts during this transition to quiescence, Lai and Doe discovered that the Prospero protein was briefly detected, at low levels, in the nucleus of the neuroblast just before it became dormant.

To see whether this 'low-level pulse' of nuclear Prospero is linked to the cell entering a dormant state, Lai and Doe investigated two types of mutant fly in which neuroblasts become dormant either earlier or later than in normal flies. A low-level pulse of nuclear Prospero still precisely matched the start of the dormant state in both mutants. When the Prospero protein was removed altogether, the neuroblasts failed to become dormant, and instead continued dividing.

Lai and Doe propose that different levels of Prospero distinguish three different fates for neuroblasts. Neuroblasts self-replicate when Prospero is kept out of the nucleus, become dormant when exposed to low level nuclear Prospero, and produce the mature cells of the brain when nuclear Prospero levels are high. Exactly how the intermediate levels of nuclear Prospero trigger the dormant state remains a question for future work, as is the question of whether the related mammalian protein, called Prox1, has a similar function.

Understanding how stem cells switch between cell division and quiescence is important for developing effective stem cell-based therapies. It could also help us understand cancer, as cancer cells go through similar periods of inactivity, during which they do not respond to many anti-tumor drugs.

in repressing cell cycle gene expression (*Dyer, 2003*; *Foskolou et al., 2013*). In addition, keeping high levels of Prospero out of the neuroblast nucleus is essential to prevent neuroblast differentiation (*Choksi et al., 2006*; *Cabernard and Doe, 2009*; *Bayraktar et al., 2010*).

In this study, we show that the Prospero differentiation factor is transiently detected at low levels in the neuroblast nucleus just prior to entry into quiescence. We find that loss of Prospero prevents entry into quiescence, whereas a pulse of low level nuclear Prospero drives proliferating larval neuroblasts into quiescence. We propose a model in which low levels of Prospero repress all cell cycle and progenitor genes except *deadpan*, whereas high levels of Prospero additionally repress *deadpan* to allow neural differentiation.

## Results and discussion

### Quiescent neuroblasts have a novel transcriptional profile lacking both progenitor and differentiation markers

To investigate the transition from a proliferating neuroblast to a quiescent neuroblast, we examined known proliferation/progenitor and differentiation marker expression during this transition. We hypothesized that loss of a progenitor marker or gain of a differentiation marker might induce neuroblast quiescence. As

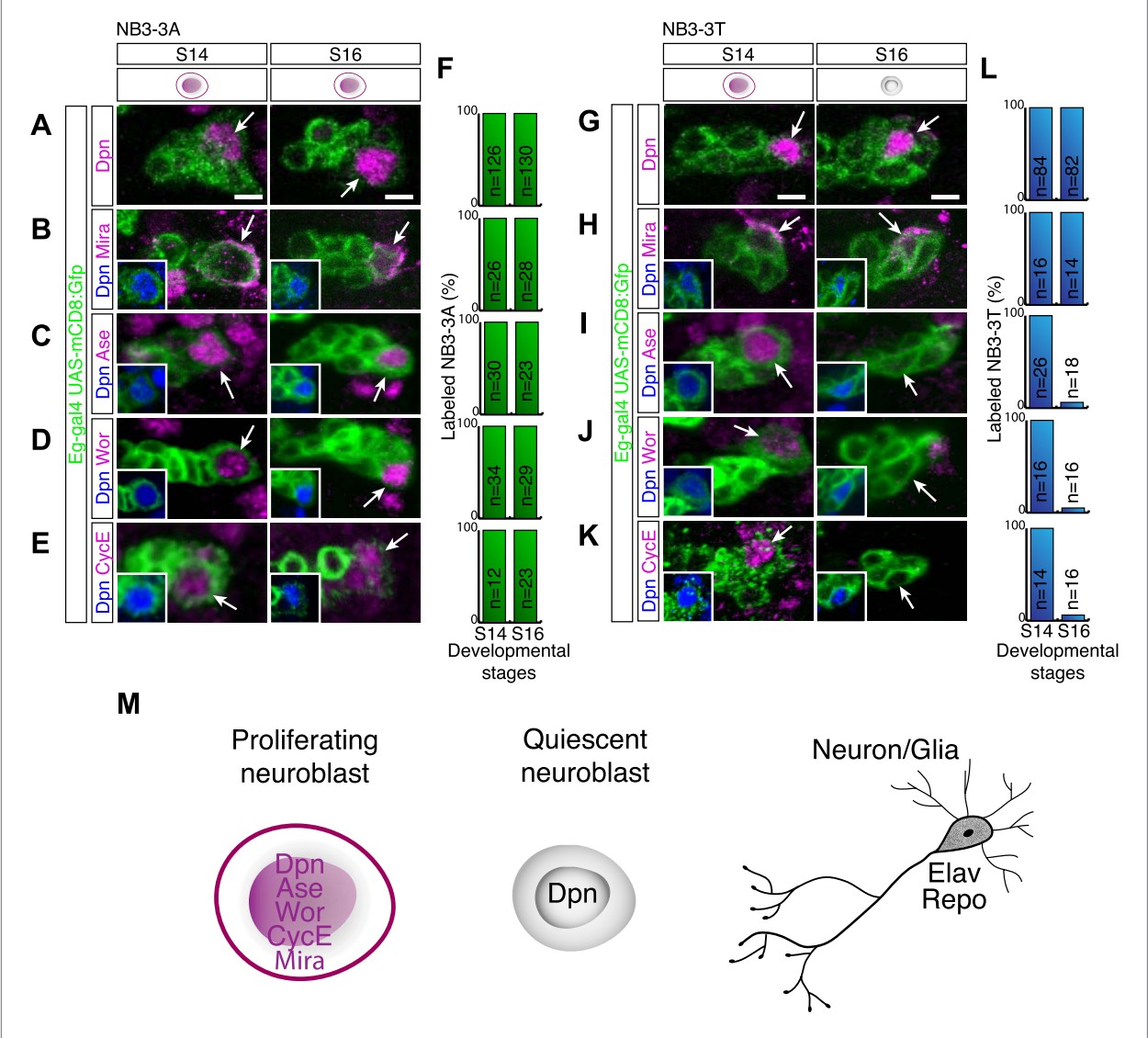

**Figure 1**. Quiescent neuroblasts have a novel transcriptional profile lacking both progenitor and differentiation markers. (**A**–**F**) Proliferating NB3-3A contains nuclear Deadpan (**A**), cytoplasmic Mira (**B**), nuclear Asense (**C**), nuclear Worniu (**D**), and nuclear CycE (**E**) at embryonic stages 14 and 16; quantification is shown in (**F**). Neuroblast lineages are marked by Eg-GFP (*Eg-gal4 UAS-mCD8:GFP*), and the neuroblast is identified by Dpn (shown in insets) and indicated by arrows. Anterior side is up, and lateral is at right. Scale bar: 5 µm. (**G**–**L**) Proliferating NB3-3T expresses all above-mentioned neuroblast markers at stage 14 (S14). At stage 16 (S16), NB3-3T is quiescent and contains Deadpan (**G**) and Mira (**H**) but lacks Asense (**I**), Worniu (**J**) and CycE (**K**). Quantified in (**L**). Scale bar: 5 µm. (**F** and **L**) Quantification; number of neuroblasts scored shown in bar. (**M**) Schematic summary of neuroblast marker profiles of proliferating neuroblast, quiescent neuroblast and neuron.

The following figure supplements are available for figure 1:

**Figure supplement 1**. Larval quiescent neuroblasts are Deadpan-positive but lack the progenitor markers Miranda, Asense, Cyclin E, Worniu and the differentiation markers Prospero, Elav, and Repo.

**Figure supplement 2**. Overexpressing Asense and Cyclin E does not change the timing of neuroblast quiescence.

a model system we used the identified neuroblast 3–3 in the thoracic segments (NB3-3T) which reliably enters quiescence at embryonic stage 15 (*Tsuji et al., 2008*). Moreover, neuroblast 3–3 in abdominal segments (NB3-3A) remains proliferative until the end of embryogenesis and provides an excellent internal control (*Tsuji et al., 2008*). Previous work showed that the quiescent NB3-3T can be recognized by Deadpan, a basic helix-loop-helix transcription factor (*Zhu et al., 2008*; *Chell and Brand, 2010*).

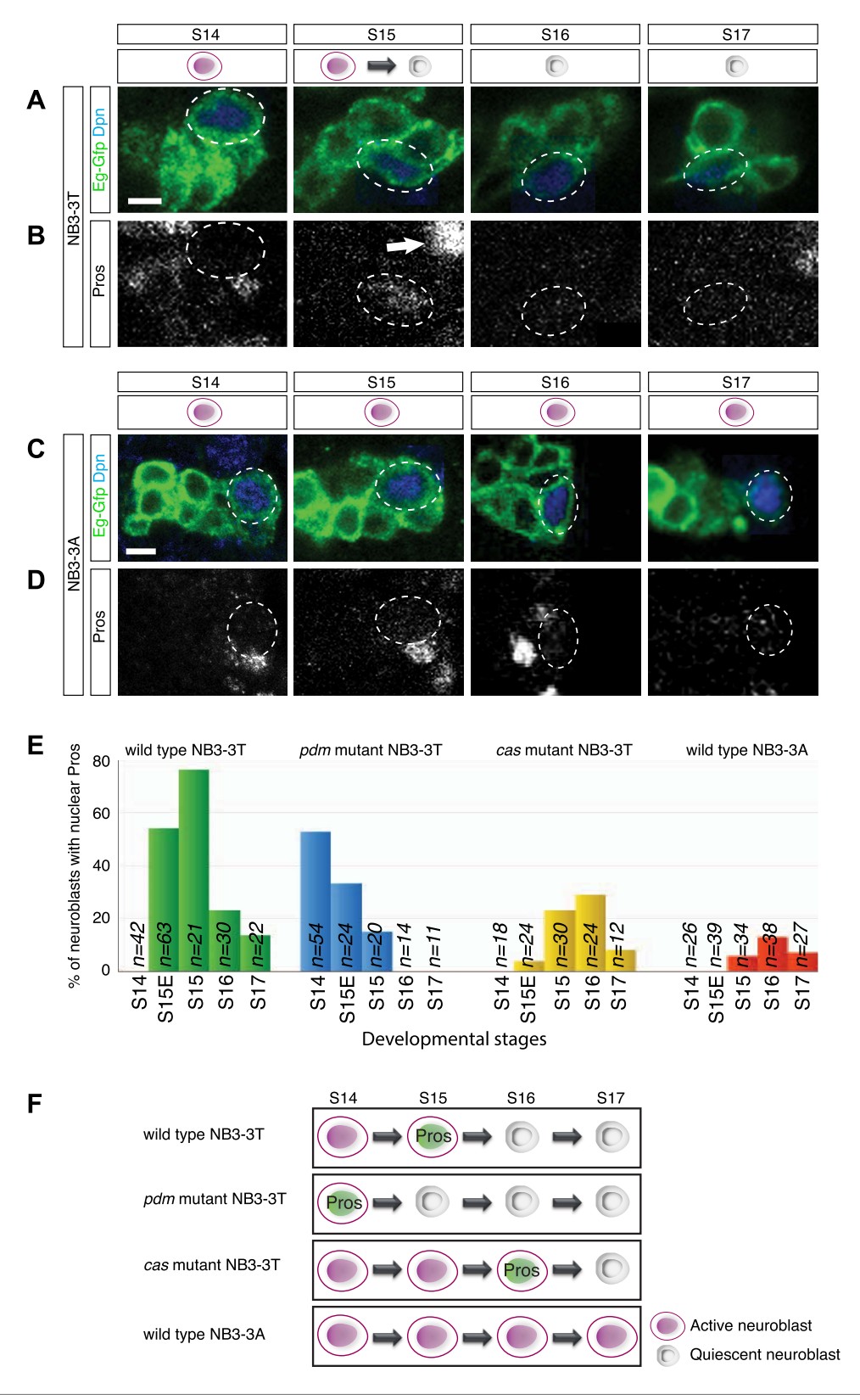

**Figure 2**. Transient low level nuclear Prospero is tightly correlated with neuroblast entry into quiescence. (**A–B**) NB3-3T shows transient nuclear Prospero (Pros) during entry into quiescence at stage 15 (S15) (**B**). Differentiated daughter cell shows strong nuclear Prospero and is indicated by the arrow. Neuroblast lineages are identified

*Figure 2. Continued*

by Eg-GFP (*Eg-gal4 UAS-mCD8:GFP*) (**A**), and the neuroblast is identified by Dpn and outlined by dashed lines. Summarized in schematic below (Prospero, green); quantified in (**E**). Anterior side is up, and lateral is at right. Scale bar: 5 µm. (**C–D**) Proliferating NB3-3A does not contain nuclear Prospero. Summarized in schematic below; quantified in (**E**). Scale bar: 5 µm. (**E**) Temporal identity factors schedule the timing of neuroblast entry into quiescence and the expression of nuclear Prospero. In *cas* mutant (*cas*[24]), neuroblast quiescence is delayed, as is the timing of nuclear Prospero. In the deficiency allele *Df(2L)ED773* which removes both *nubbin/pdm2* (*pdm* mutant), NB3-3T precociously enters quiescence, and timing of nuclear Prospero is advanced. (**F**) Schematic of the timing of neuroblast entry into quiescence and the expression of nuclear Prospero.

The following figure supplement is available for figure 2:

**Figure supplement 1**. Nuclear Prospero levels are lower in stage 15 NB3-3T than in differentiating GMCs.

The coiled-coil Miranda protein is also reported to mark quiescent neuroblasts (*Tsuji et al., 2008*), which we confirm here (*Figure 1*), but note that *mira* transcription is down-regulated in quiescent neuroblasts based lack of mRNA in the stage 17 CNS in the BDGP in situ database (*Tomancak et al., 2007*). Interestingly, we find that all other tested progenitor markers—Worniu, Asense, and Cyclin E (CycE) (*Brand et al., 1993*; *Caldwell and Datta, 1998*; *Ashraf et al., 2004*; *Lai et al., 2012*)—are not expressed in the quiescent NB3-3T (*Figure 1*; data not shown), although they can be robustly detected in the proliferative NB3-3A (*Figure 1*). Similarly, larval quiescent neuroblasts lack expression of Worniu, Asense, CycE (and Miranda), while retaining Deadpan protein and active *deadpan* transcription (*Figure 1—figure supplement 1*). However, misexpression of Worniu (data not shown), Asense, or CycE (*Figure 1—figure supplement 2*) has no effect on the timing of NB3-3T quiescence.

We next turned to examining differentiation markers. We assayed nuclear Prospero, which marks differentiating GMCs and young neurons in the larval CNS (*Carney et al., 2013*), Embryonic lethal and abnormal vision (Elav), which marks all neurons (*Robinow and White, 1988*); and Reversed polarity (Repo), which marks all non-midline glia (*Xiong et al., 1994*). We found that the mature quiescent NB3-3T did not express nuclear Prospero, Elav, or Repo (data not shown), nor were these differentiation markers observed in quiescent larval brain neuroblasts (*Figure 1—figure supplement 1*). Interestingly, we did observe transient low level nuclear Prospero in NB3-3T just prior to its entry into quiescence (see next section). We conclude that quiescent neuroblasts have a novel transcriptional profile that lacks both progenitor and differentiation markers.

## A pulse of low level nuclear Prospero marks neuroblast entry into quiescence

Here, we characterize further our observation that NB3-3T showed low level nuclear Prospero at stage 15, concurrent with the entry into quiescence. Whereas NB3-3T showed a pulse of nuclear Prospero at stage 15 concurrent with entry into quiescence (*Figure 2A,B*), NB3-3A did not show nuclear Prospero and did not enter quiescence (*Figure 2C,D*). The levels of Prospero in the NB3-3T neuroblast are clearly detectable by immunofluorescent staining, but at significantly lower levels than in differentiating GMCs (*Figure 2B*, arrow; *Figure 2—figure supplement 1*). To probe the correlation of transient nuclear Prospero and neuroblast quiescence in more detail, we assayed temporal identity mutations that are known to shift the timing of NB3-3T quiescence: *nubbin/pdm2* (*pdm*) mutants cause precocious NB3-3T quiescence, whereas *castor* (*cas*) mutants cause a delay in NB3-3T quiescence (*Tsuji et al., 2008*). Strikingly, the pulse of nuclear Prospero precisely matched the timing of neuroblast quiescence: it occurred earlier in *pdm* mutants, and later in *cas* mutants (*Figure 2E,F*). We conclude that a pulse of low level nuclear Prospero marks neuroblast entry into quiescence, suggesting a functional relationship.

## Prospero is required for neuroblast quiescence

Having shown a strong correlation between the timing of nuclear Prospero and neuroblast entry into quiescence, we next asked if Prospero is required for neuroblast entry into quiescence and loss

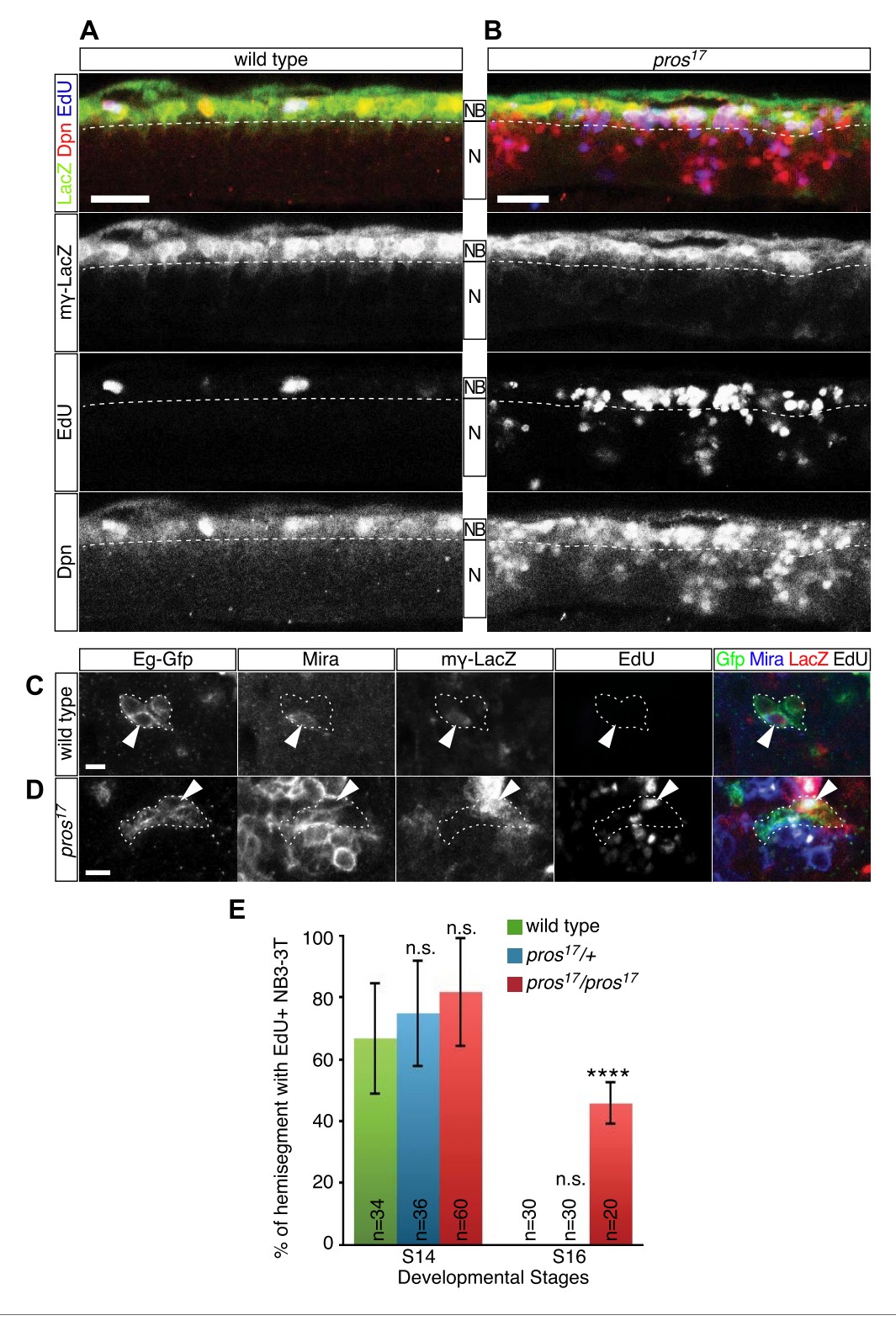

**Figure 3**. Prospero is required for neuroblast quiescence. (**A**–**B**) Wild type and *prospero* mutant (*pros^17^*) stage 16 embryos. Parental neuroblasts (NB) stain for Deadpan and mγ-LacZ, whereas ectopic 'de-differentiated' neuroblasts stain for Deadpan but not mγ-LacZ. Proliferating neuroblasts are marked by EdU incorporation. (**A**) In wild type, most thoracic neuroblasts are in quiescence and do not incorporate EdU. (**B**) In *prospero* mutants, most
*Figure 3. Continued on next page*

*Figure 3. Continued*

parental neuroblasts (mγ-LacZ+) have not entered quiescence and still incorporate EdU. Ventral side is up and anterior is at left. Dashed line, boundary of neuroblast and neuron (N) layers. Scale bars: 20 μm. (**C–E**) NB3-3T remains proliferative in *prospero* mutant stage 16 embryos. (**C**) In wild type, NB3-3T is Mira+ mγ-LacZ+ EdU−; (**D**) in *prospero* mutants, NB3-3T is Mira+ mγ-LacZ+ EdU+. The NB3-3T lineage was identified by Eg-GFP (*Eg-gal4 UAS-mCD8:GFP*) and the neuroblast is indicated by arrowheads. (**E**) Quantification. n.s., not significant; ****p < 0.00001.

of nuclear Prospero can delay the timing of neuroblast quiescence. In theory, this can be done by quantifying the number of EdU+ neuroblasts in *prospero* null mutant embryos. The problem is that *prospero* mutants show de-differentiation of GMCs into proliferating neuroblasts (*Choksi et al., 2006*; *Lee et al., 2006*), making it necessary to distinguish the parental neuroblasts from the de-differentiated GMCs. To resolve the issue, we used the Notch signaling reporter, mγ-LacZ (*Wech et al., 1999*; *Cooper et al., 2000*), to identify the parental neuroblasts. During neuroblast asymmetric cell division, Miranda-Prospero and Partner of numb (Pon)-Numb protein complexes are independently segregated into the GMC to promote differentiation (*Doe, 2008*). In *prospero* mutants, Numb is still properly segregated to GMCs where it represses Notch activity (*Spana and Doe, 1995*) and consequently the Notch reporter mγ-LacZ is restricted to parental neuroblasts. *mγ-LacZ* is transcribed only in neuroblasts; in wild type the protein perdures into neuroblast progeny, whereas in *prospero* mutants the protein is restricted to neuroblasts because the progeny proliferate without expressing mγ-LacZ and thus dilute out the protein (*Choksi et al., 2006*; *Figure 3A,B*). Thus, we scored for EdU incorporation in the mγ-LacZ+ parental neuroblasts. Whereas in wild type nearly all neuroblasts ceased EdU incorporation by stage 16 (*Figure 3A*), many mγ-LacZ+ neuroblasts continued to incorporate EdU in *prospero* mutants at the same stage (*Figure 3B*). This strongly supports a model in which Prospero is required for neuroblast entry into quiescence. In addition, Prospero is unique among basal cell fate determinants in regulating neuroblast quiescence: loss of function mutations in *numb* or *brat*, or the Prospero target gene *string*, showed no effect on the timing of neuroblast quiescence (data not shown).

To test further whether Prospero is required for neuroblast quiescence, we returned to the NB3-3T model system. We used mγ-LacZ, Miranda, and EdU incorporation to assay the timing of NB3-3T quiescence in wild type and *prospero* mutant embryos. In both wild type and *prospero* haplo-insufficiency embryos, the mγ-LacZ+ NB3-3T was proliferative at stage 14 and quiescent by stage 16 (*Figure 3C*, quantified in *Figure 3E*). In *prospero* mutant embryos, the mγ-LacZ+ NB3-3T was proliferative at both stage 14 and 16 (*Figure 3D*, quantified in *Figure 3E*). The results suggest that the level of Prospero required for neuroblast quiescence is lower than that in *prospero* haplo-insufficiency and that a low level of Prospero is sufficient to promote neuroblast quiescence (see next section). We conclude that Prospero is required for neuroblast quiescence.

## Prospero is sufficient to induce neuroblast quiescence

The central brain neuroblasts are continuously proliferating during the third larval instar (*Ito and Hotta, 1992*), and so we used this stage to determine whether transient Prospero expression could induce neuroblast quiescence. We used the TARGET method (*McGuire et al., 2003*) to transiently induce Prospero expression for 2 hr, and then assayed for neuroblast quiescence 12 hr later (*Figure 4*). Because high levels of Prospero can induce neuroblast differentiation (*Figure 4—figure supplement 1*) (*Choksi et al., 2006*; *Cabernard and Doe, 2009*; *Bayraktar et al., 2010*), we used Prospero levels low enough to minimize neuroblast differentiation, as shown by the persistence of most Deadpan+ neuroblasts after misexpression (*Figure 4D*). We identified quiescent neuroblasts by their failure to incorporate EdU, lack of progenitor marker Worniu, and lack of the differentiation marker Elav— together a robust signature for quiescent neuroblasts (see *Figure 1*). In wild-type larval brains, we found few or no quiescent neuroblasts, as expected (*Figure 4E,F*). In contrast, a pulse of Prospero resulted in the appearance of numerous quiescent neuroblasts (*Figure 4E,F*; 12 hr timepoint). To rule out the possibility that these 'quiescent neuroblasts' were early in the differentiation pathway, we reasoned that if they were quiescent they could reactivate proliferation in a nourishing environment (*Chell and Brand, 2010*; *Sousa-Nunes et al., 2010*) whereas if they were differentiating they would never re-enter the cell cycle. Thus, we exposed larval neuroblasts to the same 2 hr

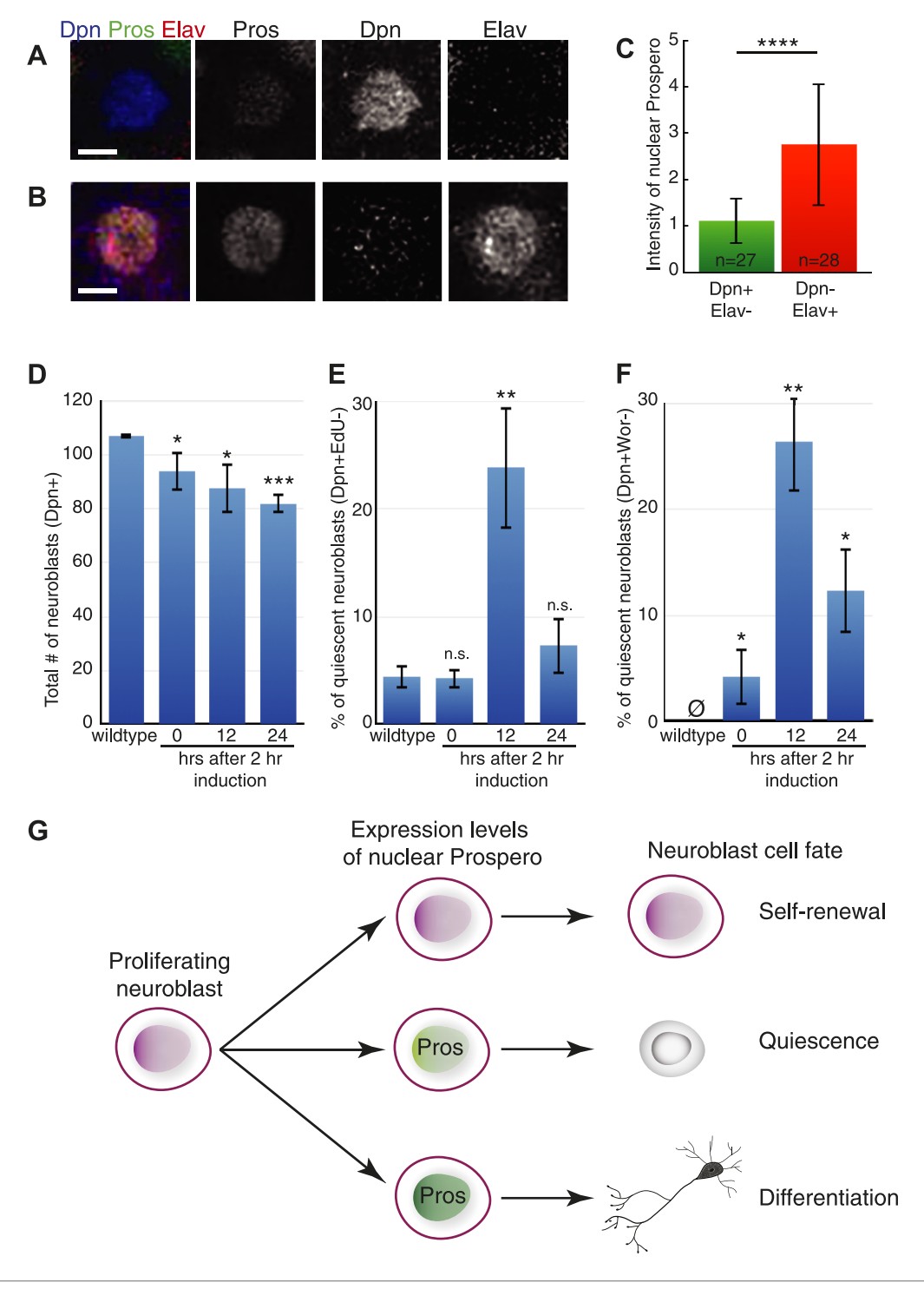

**Figure 4**. Prospero is sufficient to induce neuroblast quiescence. (**A**–**C**) Transient overexpression of nuclear Prospero in neuroblasts (*worniu-gal4 UAS-HA:prospero tub-gal80^ts*) results in neuroblast quiescence (**A**) or differentiation (**B**). Third-instar larvae were shifted from 22°C to 30°C to inactivate Gal80, which resulted in nuclear Prospero in neuroblasts. Low levels of nuclear Prospero result in neuroblast quiescence (Dpn+ Elav–) (**A**) whereas high levels of nuclear Prospero suppress Deadpan and activate Elav to induce differentiation (Dpn– Elav+) (**B**). Prospero levels were quantified in (**C**). Scale bars: 5 μm. (**C**) Quantification of Prospero levels in quiescent neuroblasts (Dpn+ Elav–) or differentiated neuroblasts (Dpn– Elav+). Differentiated neuroblasts were identified by their nuclear size larger

*Figure 4. Continued on next page*

*Figure 4. Continued*

than 7 μm in diameter. Prospero intensity was determined by totaling the gray value in nucleus of confocal stacks, followed by normalization to the total gray value of DNA marker 4',6-diamidino-2-phenylindole (DAPI). Number of neuroblasts quantified shown in bar. Error bars: standard deviation. (**D–F**) Larval neuroblast phenotype upon transient overexpression of Prospero. (**D**) Neuroblasts stay Deadpan⁺ (Dpn⁺) and thus do not differentiate. (**E**) There is an increase of quiescent neuroblasts 12 hr after Prospero induction, as determined by the elevated percentage of EdU⁻ (**D**) and Wor⁻ (**E**) neuroblasts. The quiescent neuroblasts re-enter the cell cycle and express the progenitor marker Wor 24 hr after induction. Error bars: standard deviation. n.s., not significant; *p < 0.05; **p < 0.01; ***p < 0.0001. (**G**) Model. Prospero levels distinguish three progenitor fates: absent for self-renewal, low for quiescence, and high for differentiation.

The following figure supplements are available for figure 4:

**Figure supplement 1**. Low nuclear Prospero induces neuroblast quiescence, whereas high Prospero induces neuroblast differentiation.

**Figure supplement 2**. Prospero is sufficient to induce embryonic neuroblast quiescence.

---

pulse of Prospero, but waited 24 hr to assay neuroblasts. We found that nearly all of the neuroblasts that were quiescent at 12 hr after induction had reactivated proliferation and re-expressed progenitor markers (***Figure 4E,F***; 24 hr timepoint). In addition, precocious low level nuclear Prospero could advance the timing of NB3-3T quiescence (***Figure 4—figure supplement 2***). We conclude that transient, low-level nuclear Prospero is sufficient to induce neuroblast quiescence. Interestingly, Prospero can also transiently arrest the cell cycle without inducing differentiation in embryonic longitudinal glial progenitors (***Griffiths and Hidalgo, 2004***), although the relative levels of Prospero have not been explored in this system.

Our results confirm that high levels of Prospero can trigger differentiation and reveal that low levels of Prospero induce quiescence in the *Drosophila* neuroblasts. High levels of the vertebrate ortholog Prox1 can also trigger differentiation (***Dyer, 2003***; ***Takahashi et al., 2006***; ***Foskolou et al., 2013***); thus it is interesting that low levels of nuclear Prospero promote neuroblast quiescence without triggering differentiation (***Figure 4—figure supplement 1***). Prox1 activates some target genes at high levels but represses some target genes at low levels (***Chen et al., 2008a***, ***2008b***). We suggest a speculative model for Prospero function: low levels of Prospero are sufficient to repress most progenitor-specific genes in neuroblasts (*worniu*, *asense*, *miranda*, and *cycE*) but do not repress the expression of *deadpan* (perhaps due to lack of high affinity Prospero binding sites). In contrast, high levels of Prospero are sufficient to repress *deadpan* (perhaps via low affinity binding sites) and may even activate neuronal differentiation genes (***Figure 4G***).

In embryonic neuroblasts, the temporal transcription factors Pdm and Cas schedule the timing of neuroblast quiescence, but how they regulate Prospero nuclear import is unknown. In larval neuroblasts, Grh prevents accumulation of nuclear Prospero which would induce neuroblast cell cycle exit and differentiation (***Maurange et al., 2008***; ***Chai et al., 2013***), again by an unknown mechanism. The protein Caliban is known to promote Prospero nuclear export in S2 cells (***Bi et al., 2005***), but it is unknown whether Caliban has a similar role in neuroblasts or if down-regulation of Caliban levels or activity leads to increased nuclear Prospero in neuroblasts entering quiescence.

## Materials and methods

### Fly genetics

The following flies were used in this study: (1) *UAS-mCD8:Gfp;eagle-gal4*; (2) *cas²⁴/TM3 Sb Ubx-lacZ*; (3) *Df(2L)ED773/CyO wg-lacZ*; (4) *mγ-lacZ;UAS-mCD8:Gfp; eg-gal4 pros¹⁷/TM3 Sb Ubx-lacZ*; (5) *UAS-HA:prospero* (attP40); (5) *UAS-HA:prospero* (attP2); (6) *wor-gal4;tubulin-gal80ᵗˢ*.

### Induction of Prospero, EdU incorporation, antibody staining and imaging

Embryos with genotype *UAS-HA:prospero/wor-gal4;tubulin-gal80ᵗˢ/+* were collected for 4 hr at 22°C, and then cultured at 22°C for 5 days. The larvae were transferred to 30°C to inactivate Gal80, which resulted in the activation of Gal4 to express HA:Prospero. After heat shock, the larvae were immediately dissected

or moved back to 22°C to incubate for 12 or 24 hr for later dissection, followed by 2-hr EdU incorporation, fixation, and antibody staining. EdU incorporation was performed in PBS at the concentration of 200 µg/ml. Antibody staining and EdU detection were previously described (*Lai et al., 2012*). Antibodies used in this study included: rabbit anti-Asense (1:1000; Cheng-Yu Lee, University of Michigan, Ann Arbor, MI), mouse anti-β-galactosidase (1:1000; Promega, Madison, WA), mouse anti-Cyclin E (1:50; Developmental Studies Hybridoma Bank (DHSB) developed under NICHD and maintained by University of Iowa), rabbit anti-Cyclin E (1:300; Santa Cruz Biotechnology, Dallas, TX); rat anti-Deadpan monoclonal (1:50; Doe lab), guinea pig anti-Deadpan (1:1000; Jim Skeath, Washington Univ., St. Louis, MO), mouse anti-Eagle (1:100; Doe lab); rabbit anti-Eagle (1:500) (*Higashijima et al., 1996*; *Freeman and Doe, 2001*); rat anti-Elav (1:50, DSHB); chicken anti-GFP (1:500; Aves Labs, Tigard, OR), mouse anti-HA (1:1000; Covance, Princeton, NJ), chicken anti-HA (1:1000, Bethyl Laboratories, Montgomery, TX); guinea pig anti-Miranda (1:2000; Doe lab), mouse anti-Prospero monoclonal purified IgG (1:1000; Doe lab), and rat anti-Worniu (1:2; Doe lab). Microscopy was done using a Zeiss LSM700 or LSM710; image processing and quantification were performed with the open source software FIJI.

## Acknowledgements
We thank Travis Carney and Minoree Kohwi for critical comments on the manuscript. We also thank Jim Skeath, Cheng-Yu Lee, and Sarah Bray for reagents. The work was supported by Howard Hughes Medical Institute, where CQD is an Investigator.

## Additional information

### Funding

| Funder | Author |
|---|---|
| Howard Hughes Medical Institute | Sen-Lin Lai, Chris Q Doe |

The funder had no role in study design, data collection and interpretation, or the decision to submit the work for publication.

### Author contributions
S-LL, Conception and design, Acquisition of data, Analysis and interpretation of data, Drafting or revising the article, Contributed unpublished essential data or reagents; CQD, Conception and design, Analysis and interpretation of data, Drafting or revising the article

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
