## [Decision Letter]

Thank you for sending your work entitled “Transient nuclear Prospero induces neural progenitor quiescence” for consideration at *eLife.* Your article has been favorably evaluated by K (Vijay) VijayRaghavan (Senior editor), a Reviewing editor, and 3 reviewers.

The Reviewing editor and the other reviewers discussed their comments before we reached this decision, and the Reviewing editor has assembled the following comments to help you prepare a revised submission.

The current study addresses the role of the Prospero transcription factor in inducing neuroblast quiescence in the embryonic and larval *Drosophila* CNS. They find that the NB3-3T lineage displays a brief pulse of weak Pros nuclear expression just prior to entering quiescence. Mutant and mis-expression analysis lends support for the notion that this brief pulse in fact induces quiescence. They also address this notion in the postembryonic neuroblasts, and find support for a similar role for Pros in quiescence. Since we know very little about the fascinating, and even potentially medically important phenomena of stem cell quiescence, this study is of importance, and would likely be interesting for a broader audience. However, the manuscript is not without problems, and several issues should be addressed prior to publication.

Substantive concerns:

1) If a weak and brief pulse of Pros protein in NB nuclei is indeed important for NB quiescence, it is possible that pros may show haplo-insufficiency. To this end, they could analyze NB3-3T quiescence in pros heterozygotes. In addition, given the role of *grh* in quiescence, any heterozygotic effect of pros may be enhanced in transhet with *grh* i.e., *pros+/-; grh+/-*. Any such genetic dosage effect would greatly strengthen the study.

2) In the Introduction section and Figure 1—figure supplement 1: They state that “However, loss or misexpression of Worniu, Asense, or CycE has no effect on the timing of NB3-3T quiescence (Figure 1—figure supplement 1; data not shown)”. I cannot find any data regarding that misexpression of these factors cannot affect the timing of NB3-3T quiescence. This is an important point in the paper, and some data should be provided.

3) In pros mutants GMCs are de-differentiated into NBs, and hence express NB markers, e.g. Dpn. Hence, they use mgamma-LacZ to identify NBs in pros mutants. This is a reasonable approach, given that the asymmetric localization of Numb into GMCs (which is not affected in pros) results in expression of mgamma-lacZ only in the NBs. However, given that b-gal has a half-life in embryos of 8-12 hrs, it is not clear to me how the b-gal can label only NBs. Yes, the transgene is only transcriptionally active in NBs, but surely the b-gal protein, being quite stable and not subjected to asymmetric distribution, must label daughter cells as well. We need to be convinced, in this context, that all or in fact any of the ectopically EdU labelled cells in Figure 3 are actually NBs. Similarly, in Figure 3, in pros mutants there is b-gal expression (as well as Mira expression) in several cells in the NB3-3T lineage; how can the authors be sure that the EdU positive cell is the NB?

4) Figure 4—figure supplement 1 and Figure 4—figure supplement 2: They mis-express pros in NB3-3T, and find that it can induce NB quiescence. This is not surprising, given the role of pros in terminating proliferation.

5) Regarding postembryonic NBs, there are at least two previous publications that have addressed the function of pros in postembryonic NBs. In the paper from the Gould lab (Maraunge et al, 2008, Supplement Figure 3) they show ectopic dividing cells in pros MARCM clones. However, since the clones are quite large, and they use few markers, ectopically dividing cells may just as well be GMCs and not NBs. Similarly, in the recent paper from the Doe lab itself (Bayraktar et al., Neural Dev, 2010,) they only study GOF effects of Pros, and find no pros LOF effects in Type I NBs. In the current study, they again show that GOF of pros can trigger NB quiescence in postembryonic NBs. This is not surprising given the two previous publications. However, they provide no new LOF data supporting the idea that Pros is necessary for NB quiescence.

6) The targets of Pros that induce/repress neuroblast quiescence are unknown. Pros has been shown to repress neuroblast cell-fate genes and cell-cycle genes in GMCs (Choksi SP, et al., Dev Cell 2006). The authors mentioned that loss or misexpression of Wor, Ase or CycE has no effect on the timing of NB3-3T quiescence. Have they tested these in 3rd instar central larval brain neuroblasts, in which the transient Pros expression was shown to induce neuroblast quiescence? Also, have they tested the role of String, an important cell cycle regulator that was repressed by Pros (Choksi SP, et al., Dev Cell 2006)?

7) The authors could also discuss the potential role of other cell fate determinants (Numb and Brat) in neuroblast quiescence. It will be valuable to know if the role of Pros represents a general function of cell fate determinants in regulating neuroblast quiescence or is Pros unique among the cell fate determinants in this process.

---

## [Author Response]

*1) If a weak and brief pulse of Pros protein in NB nuclei is indeed important for NB quiescence, it is possible that pros may show haplo-insufficiency. To this end, they could analyze NB3-3T quiescence in pros heterozygotes. In addition, given the role of* grh *in quiescence, any heterozygotic effect of pros may be enhanced in transhet with* grh *i.e.,* pros+/-; grh+/-*. Any such genetic dosage effect would greatly strengthen the study*.

We appreciate reviewers raising the point and help us to further strengthen our conclusion: “low levels of Prospero promote neuroblast quiescence.” We first performed EdU incorporation in the *pros* heterozygous mutants and found that the neuroblast quiescence timing was not delayed. The results suggest that the levels of Prospero required for neuroblast quiescence is lower than that in *pros* haplo-insufficiency, which is consistent with our conclusions from the overexpression studies. We have added the result in Figure 3 and revised the Results section: “In both wild type and prospero haplo-insufficiency embryos, the mγ-LacZ+ NB3-3T was proliferative at stage 14 and quiescent by stage 16 (Figure 3, quantified in 3E). In prospero mutant embryos, the mγ-LacZ+ NB3-3T was proliferative at both stage 14 and 16 (Figure 3, quantified in 3E). The results suggest that the level of Prospero required for neuroblast quiescence is lower than that in prospero haplo-insufficiency and that a low level of Prospero is sufficient to promote neuroblast quiescence.”

Although *grh* plays an important role in regulating Pros activity in larval neuroblasts (35; 13), it is unknown whether *grh* plays a similar role in embryonic neuroblasts. If *grh* regulates the timing of neuroblast quiescence by preventing Pronuclear accumulation, trans-heterozygous *pros+/- grh+/-* will show no phenotype as Grh and Pros counteract each other in regulating neuroblast quiescence. As expected, we didn’t find any phenotype regarding the timing of neuroblast quiescence in the *pros+/- grh+/-*trans-heterozygous mutants (data not shown). Nevertheless, the concern led us to put back the discussion we removed in the previous submission due to the limitation of space: “In embryonic neuroblasts, the temporal transcription factors Pdm and Cas schedule the timing of neuroblast quiescence, but how they regulate Prospero nuclear import is unknown. In larval neuroblasts, Grh prevents accumulation of nuclear Prospero which would induce neuroblast cell cycle exit and differentiation (13; 35), again by an unknown mechanism.”

*2) In the Introduction section and*
Figure 1—figure supplement 1*: They state that “However, loss or misexpression of Worniu, Asense, or CycE has no effect on the timing of NB3-3T quiescence (*Figure 1—figure supplement 1*; data not shown)”. I cannot find any data regarding that misexpression of these factors cannot affect the timing of NB3-3T quiescence. This is an important point in the paper, and some data should be provided*.

We agree, and we have added a new figure with the requested data (Figure 1—figure supplement 2). In summary, when we overexpressed Ase or CycE, we didn’t observe significant change in the number of proliferating neuroblasts.

*3) In pros mutants GMCs are de-differentiated into NBs, and hence express NB markers, e.g. Dpn. Hence, they use mgamma-LacZ to identify NBs in pros mutants. This is a reasonable approach, given that the asymmetric localization of Numb into GMCs (which is not affected in pros) results in expression of mgamma-lacZ only in the NBs. However, given that b-gal has a half-life in embryos of 8-12 hrs, it is not clear to me how the b-gal can label only NBs. Yes, the transgene is only transcriptionally active in NBs, but surely the b-gal protein, being quite stable and not subjected to asymmetric distribution, must label daughter cells as well. We need to be convinced, in this context, that all or in fact any of the ectopically EdU labelled cells in*
Figure 3
*are actually NBs. Similarly, in*
Figure 3*, in pros mutants there is b-gal expression (as well as Mira expression) in several cells in the NB3-3T lineage; how can the authors be sure that the EdU positive cell is the NB?*

We thank the reviewers for highlighting this issue, which obviously needed a better explanation! The short answer is that *pros* mutant GMCs do not transcribe mγ-lacZ but continue to divide and thus dilute out mγ−lacZ, resulting in high level mγ−lacZ specifically in the parental neuroblasts. We have clarified this in the Results section: “mγ-LacZ is transcribed only in neuroblasts; in wild type the protein perdures into neuroblast progeny, whereas in prospero mutants the protein is restricted to neuroblasts because the progeny proliferate without expressing mγ-LacZ and thus dilute out the protein (Choksi et al.; Figure 3).”

We thank the reviewers for helping us clarify this important point, which further supports our identification of NB3-3T in the *prospero* mutants, and our conclusion that Prospero is required for neuroblast quiescence.

*4)*
Figure 4—figure supplement 1 and Figure 4—figure supplement 2*: They mis-express pros in NB3-3T, and find that it can induce NB quiescence. This is not surprising, given the role of pros in terminating proliferation*.

Yes, we agree that a major role of Prospero in neural progenitor cells is to terminate proliferation as previously reported (34; 17). However, we want to emphasize that our new discovery is the dosage effect of Prospero in the progenitor cells: high levels of Pros will terminate proliferation and induce differentiation, while low levels of Pros transiently arrest cell cycle but are not sufficient to change the neuroblast identity, which leads to quiescence. We have added the following sentence in the Result section to further emphasize our main conclusion in the current study: “Our results confirm that high levels of Prospero can trigger differentiation, and reveal that low levels of Prospero induce quiescence in the *Drosophila* neuroblasts.”

*5) Regarding postembryonic NBs, there are at least two previous publications that have addressed the function of pros in postembryonic NBs. In the paper from the Gould lab (**Maraunge et al, 2008**, Supplement*
Figure 3*) they show ectopic dividing cells in pros MARCM clones. However, since the clones are quite large, and they use few markers, ectopically dividing cells may just as well be GMCs and not NBs. Similarly, in the recent paper from the Doe lab itself (Bayraktar et al., Neural*
*Dev, 2010**,) they only study GOF effects of Pros, and find no pros LOF effects in Type I NBs. In the current study, they again show that GOF of pros can trigger NB quiescence in postembryonic NBs. This is not surprising given the two previous publications. However, they provide no new LOF data supporting the idea that Pros is necessary for NB quiescence*.

In our Results section, we have shown that loss of Prospero can delay the timing of neuroblast quiescence (Figure 3). We have also shown that transient overexpression of low levels of Prospero can induce neuroblast quiescence (Figure 4). To help readers easily understand our studies, we have revised the first sentence in the section of “Prospero is required for neuroblast quiescence” as follows: “Having shown a strong correlation between the timing of nuclear Prospero and neuroblast entry into quiescence, we next asked if Prospero is required for neuroblast entry into quiescence and loss of nuclear Prospero can delay the timing of neuroblast quiescence.”

*6) The targets of Pros that induce/repress neuroblast quiescence are unknown. Pros has been shown to repress neuroblast cell-fate genes and cell-cycle genes in GMCs (Choksi SP, et al.,*
*Dev Cell 2006**). The authors mentioned that loss or misexpression of Wor, Ase or CycE has no effect on the timing of NB3-3T quiescence. Have they tested these in 3rd instar central larval brain neuroblasts, in which the transient Pros expression was shown to induce neuroblast quiescence? Also, have they tested the role of String, an important cell cycle regulator that was repressed by Pros (Choksi SP, et al.,*
*Dev Cell 2006**)?*

In our earlier paper ([32], Dev Cell), we showed that *wor* mutant larval neuroblasts prematurely differentiate or die, while overexpression of Wor results in the cell cycle arrest but not quiescence. We have also used *cycE-RNAi* (FBti0140127) and *ase-RNAi* (FBti0158728) to knock down CycE and Ase, respectively, but could not induce larval neuroblast quiescence (data not shown). We thus conclude that loss of CycE, Ase or Wor do not induce larval neuroblast quiescence.

As requested, we tested the role of String in neuroblast quiescence. We examined stg mutants for neuroblast EdU incorporation and Ase, Wor and Mira markers all are absent in quiescent neuroblasts. Interestingly, *string* mutant neuroblasts are EdU- as expected, but did not downregulate Ase, Wor and Mira, showing that they did not match the gene expression profile of quiescent neuroblasts. We conclude that although *string* is repressed by Pros, *string* downregulation is not sufficient to induce neuroblast quiescence. We add: “In addition, Prospero is unique among basal cell fate determinants in regulating neuroblast quiescence: loss of function mutations in numb or brat, or the Prospero target gene string, showed no effect on the timing of neuroblast quiescence (data not shown)”.

*7) The authors could also discuss the potential role of other cell fate determinants (Numb and Brat) in neuroblast quiescence. It will be valuable to know if the role of Pros represents a general function of cell fate determinants in regulating neuroblast quiescence or is Pros unique among the cell fate determinants in this process*.

We thank the reviewers for this comment, which allowed us to add new experiments that further highlight the unique role of Pros in promoting neuroblast quiescence. We found that *numb* or *brat* null mutant embryos did not exhibit a delay or advance in neuroblast quiescence (data not shown). We add: “In addition, Prospero is unique among basal cell fate determinants in regulating neuroblast quiescence: loss of function mutations in numb or brat, or the Prospero target gene string, showed no effect on the timing of neuroblast quiescence (data not shown)”.